# Design and Implementation of Morphed Multi-Rotor Vehicles with Real-Time Obstacle Detection and Sensing System

**DOI:** 10.3390/s21186192

**Published:** 2021-09-15

**Authors:** Aleligne Yohannes Shiferaw, Balasubramanian Esakki, Tamilarasan Pari, Elangovan Elumalai, Saleh Mobayen, Andrzej Bartoszewicz

**Affiliations:** 1Department of Mechanical Engineering, Centre for Autonomous System Research, Vel Tech Rangarajan Dr Sagunthala R & D Institute of Science and Technology, Avadi, Chennai 600062, India; yohannesaleligne@gmail.com; 2Department of Aeronautical Engineering, Centre for Autonomous System Research, Vel Tech Rangarajan Dr Sagunthala R & D Institute of Science and Technology, Avadi, Chennai 600062, India; tamilarasan1837@gmail.com; 3Department of Electronics and Communication Engineering, Centre for Autonomous System Research, Vel Tech Rangarajan Dr Sagunthala R & D Institute of Science and Technology, Avadi, Chennai 600062, India; elangovan27sri@gmail.com; 4Future Technology Research Center, National Yunlin University of Science and Technology, 123 University Road, Section 3, Douliou 64002, Taiwan; 5Institute of Automatic Control, Lodz University of Technology, 18/22 Stefanowskiego St., 90-924 Lodz, Poland; andrzej.bartoszewicz@p.lodz.pl

**Keywords:** foldable UAV, morphology, kinematics, synchronous motion, obstacle detection, ultrasonic sensor

## Abstract

Multirotor unmanned aerial vehicles (MUAVs) are becoming more prominent for diverse real-world applications due to their inherent hovering ability, swift manoeuvring and vertical take-off landing capabilities. Nonetheless, to be entirely applicable for various obstacle prone environments, the conventional MUAVs may not be able to change their configuration depending on the available space and perform designated missions. It necessitates the morphing phenomenon of MUAVS, wherein it can alter their geometric structure autonomously. This article presents the development of a morphed MUAV based on a simple rotary actuation mechanism capable of driving each arm’s smoothly and satisfying the necessary reduction in workspace volume to navigate in the obstacle prone regions. The mathematical modelling for the folding mechanism was formulated, and corresponding kinematic analysis was performed to understand the synchronous motion characteristics of the arms during the folding of arms. Experiments were conducted by precisely actuating the servo motors based on the proximity ultrasonic sensor data to avoid the obstacle for achieving effective morphing of MUAV. The flight tests were conducted to estimate the endurance and attain a change in morphology of MUAV from “*X*-Configuration” to “*H*-Configuration” with the four arms actuated synchronously without time delay.

## 1. Introduction

In recent times, MUAVs have been widely used in multifaceted environments and applications, including agriculture, transportation, security, entertainment, search and rescue missions, as well as for unmanned missions in water quality monitoring and assessment using (Internet of things) IoT based Sensors [1,2,3,4]. The inherent characteristics of MUAVs, such as swift manoeuvrability and hovering abilities, are added advantages for efficient navigation to inspect collapsed buildings and even examine underground tunnels. The medium and larger-sized MUAVs have a greater load-bearing capacity and can perform various designated tasks effectively. However, their portability and manoeuvrability have become a severe issue. Thus, to be entirely applicable for various applications, having a low workspace volume is vital. Nonetheless, the current MUAV cannot still adopt into confined and narrow regions, and also, the tasks commonly observed in birds may not be possible with MUAV [5]. The flying birds are naturally adapting to various flying conditions [6]; for example, pigeons fold their wings and perform swift movements to optimize their drifting efficiency over a broader range of speeds [7,8]. In general, pigeons are changing their morphology by swapping their wings to traverse through a narrow gap. They can fold their wings tight and close to their body to perform navigation in the confined regions [9]. MUAVs should be competent enough for recognizing changes in the environment and accordingly morph into the desired shape for achieving effective navigation as birds do [10]. In this way, MUAV can be flown in narrow gaps without miniaturizing their size, which is advantageous in achieving more endurance and payload. Similarly, a large size MUAV can change its size and shape by folding its arms for navigating into a narrow space [11]. This desideratum calls for a new research topic called morphological MUAVs with a unique self-harm management feature [12,13]. The morphing of MUAV is achieved by folding their arms concerning the space constraints during their flight conditions. They can navigate effectively in obstacle-prone regions without any difficulties such as birds. 

Earlier studies on changing the relative orientation of the propellers [14] or leaning rotors were conducted to enhance the platform’s controllability [15,16]. Even though these approaches are advantageous to expedite navigation and perform manipulation tasks effectively, they may not change the shape of MUAV to reduce the work volume. There are two approaches to achieve the morphing of MUAVs. The first approach enables changing the orientation of MUAVs during flight. However, the second approach can only change its orientation before take-off or after landing. Tuna et al. [13] designed and developed a FOLLY: that can fold and unfold its arm autonomously using a crank-rocker mechanism, and it is capable of switching its orientation before or after a flight. However, their mechanism’s design did not consider a change in morphology during the flight when they face obstacles and narrow gaps. Other authors also investigate MUAVs that can change their configuration during flight to surmount through narrow gaps or grasp objects. For example, in [11], they implemented a straightforward approach using a wire-based mechanism. It enables the Quad-morphing robot to preserve structural simplicity, but it exhibits high speed and large space when it changes its configuration. A morphing Quadrotor developed in [17] can alter its configuration to envelop objects and grasp them. The mechanical design contains many components that increased the complexity of the structure. In [18], simulation analysis was performed for achieving change in morphology of MUAV wherein it can swirl and contract its arms, but they cannot handle asymmetrical configurations. Zhao et al. [19] designed a novel deformable MUAV actuated by a scissor-like morphing mechanism. Their design reduces the volume of the MUAV and enables a symmetrical deformation of the whole-body frame. Mintchev et al. [20] developed a miniaturized portable MUAV with foldable arms. It can be encased for transportation by folding its arms around the mainframe. They designed lightweight and self-deployable arms for small MUAV by using an origami technique. 

Similarly, Yang et al. [21] demonstrated a new foldable MUAV constructed with an origami approach. They have used a multi-layered manufacturing technique to produce bending-based hinges that can be depicted as revolute joints, allowing the arms to be flexible. It can fold its arms using an extra actuator and unfold it by spring effect due to the torque induced by propellers during flight to enable aggressive turning manoeuvres and operations in congested circumstances. The MUAV frame was built from laser-cut cardboard, which significantly reduced costs and shortened production time. Despite being a lightweight, self-deployable and foldable MUAV, it had limited stiffness and could not withstand greater deformation. Furthermore, the structure’s endurance and possible vibrations from the motors on the frame material limit its practical application in a real-world environment. Hintz et al. [22] presented a novel rotary-wing MUAV design with morphing capabilities. They have suggested a multicopter platform of “H-type” configuration that can spin its body into a vertical flight mode. The vehicle incorporates a rotary mechanism at the end of each arm that links the rotors, allowing the multirotor’s body to be positioned vertically or horizontally. The vertical flight mode allows navigation in small regions that are inaccessible to typical multicopter platforms flying horizontally. The morphing system makes the multi-copter resistant to rotor failures since it may spin the propellers in any direction while leaving the functional rotors at the top. Bucki and Mueller [23] developed a MUAV with rotary joints to achieve swift aerial morphing without additional actuators. The spring hinge connection between the arm and the central body reduces the vehicle’s largest dimension by approximately 50%. The vehicle design differs from that of normal MUAVs in which it has a passive hinge where the arms can engage into the central body and the use of constant force springs to draw the arms close to one another. The position of the hinge was chosen such that the reconfigured vehicle reduced its size as tiny as possible. The vehicle’s capacity to shrink its size while in flight allows it to fly through narrow gaps that a non-morphing MUAV would not be able to move through. With soft technologies and innovative manufacturing methods, Mintchev and Floreano [20] established an adaptable morphology design of MUAV that enhances its flexibility and efficiency. The wings were composed of a flexible carbon fibre skeleton wrapped with a rubber elastomeric membrane. The developed MUAV was capable of deploying its arms autonomously in response to turbulence by extending the membrane. Xiu et al. [24] proposed the design of autonomous folding and the deployment of a mechanism for MUAVs that would allow the system to react swiftly in congested environments. The reconfigurable rotor-arm system comprises primarily of a platform, rotor arms and motor-propeller sets. The motor and gear system integrated into the deployable carrier controls the movement of the rotor arms. In contrast to typical MUAVs with fixed arms, the distance between two opposing rotors can be changed by switching the location of the rotor arms. Wallace [18] created a MUAV with dynamic geometrical morphing that can mechanically change its arm’s length and angle. They methodically build the dynamic equations that control vehicle geometric morphing. The ability to fly the vehicle with a single throttle channel and the roll and pitch movement of the MUAV via geometric morphing were validated in simulation. Falanga et al. [12] developed and tested multiple adaptive morphological MUAVs, including “O,” “T,” and “H,” configurations by controlling each of the four arms linked to servo motors individually. The morphing method comprises two interconnected elements: a frame with four independently rotating arms that fold around the mainframe and a control scheme capable of accounting for the vehicle’s present morphology and ensuring stable flight at all times. The adjacent motors are vertically offset from one another to prevent propellers from clashing.

Yang et al. [25] presented a novel scalable multicopter whose deformation mechanism was a combination of “simple non-intersecting angulated elements and straight scissor-like elements (SNIAE-SSEs)”. They proved that the SNIAE-SSEs mechanism could allow one degree-of-freedom (DOF) faster actuation with a large deformation ratio, but they did not consider autonomous and controllable deformations of the arm. Tothong et al. [26] presented two versions of morphing MUAVs. First version utilizes servo motors on each propeller arm to realize a folding motion, while the second version employed micro linear actuators to scale up and down the MUAV in all directions. 

Balasubramanian et al. [27] developed a lightweight MUAV structure through the synergistic application of design analysis and fused deposition modelling. Experimental results proved that the developed MUAV has low weight and smaller parts than the conventional MUAVs, but it lacks the feature to adjust its shape and size depending on the surrounding for narrow and confined environmental conditions.

Zhao et al. [28] demonstrated inertia variations due to adaptive morphology that influenced MUAV flying performance. Extensive computational and experimental assessments were carried out and investigated on a custom-built morphing MUAV in the presence of persistent wind disturbance. The simulation findings show that the smaller MUAV has superior agility performance due to its smaller volume/size and lower weight, but the larger one looks to have stronger flying resilience in a disturbed environment. Experimental results further prove that by flexibly altering the volume/size of MUAVs with the bioinspired in-flight morphing behaviour, which was possible to execute the path following tasks in a confined space without straying the trajectory in the obstacle’s prone region. Mangersnes et al. [29] developed a drone called “LisHawk drone”. They apply the avian-based morphing strategy which generates thrust by flapping its wings to reinforce agility, manoeuvrability, inherent stability, flight speed range and power requirement. The morphing mechanism was composed of a relatively long tail and short wings with a large wing chord that enable large geometrical changes to efficiently adapt between different flight conditions. Usherwood et al. [30] developed morphing MUAV wings by converting wing principles discovered in birds into mechanical designs using a technique known as bioinspiration or biomimetics. This bio-hybrid method enables using materials in robots that are too difficult to manufacture and do not yet completely understand, such as flying feathers. They combined avionics, biomechanics, aerial robotics and biomaterials to discover how birds achieve wing morphing and implemented it in MUAVs that mimic birds. Experimental results show that the avian-inspired, morphing drone can swiftly pitch up, work at high angles of attack, and attain sturdy trim flight compared with fixed-wing drones. Chang et al. [31] developed a drone using a soft biohybrid morphing wing with feathers. The avian-inspired underactuated formation has a lower weight, and the softness of the morphing veneer provide mechanical robustness. Ajanic et al. [29] presented bioinspired wing and tail morphing to improve drone flight capabilities and verified that the evolved morphing strategy can improve agility, manoeuvrability, stability and flight speed range, and the required power of drone. 

Fabris et al. [32] investigated the consequences of the partial overlap between the propellers and the main body of a morphing MUAVs. Experiments were conducted to show the corollary, and a morphology-aware control scheme was developed. They also created a correction method to compensate for the morphing MUAVs aerodynamic impacts. The same tests were conducted and compared during flight by activating and deactivating the compensation method to ensure the correct trade-off between efficiency and vehicle compactness. Kose and Oktay [33] used the simultaneous perturbation stochastic approximation (SPSA) optimization technique to create MUAVs with collective morphing. The morphing behaviour and optimum PID coefficients were identified using state-space model methods to improve the MUAVs longitudinal and lateral flight stability during morphing.

The aforementioned works can mitigate the issues pertaining to increase the versatility of MUAVs by adjusting their shape to perform diverse missions while obstructing manoeuvrability compromises. Nevertheless, those mechanisms require more extended time for altering the configuration; thus, complex manoeuvres will be executed with a considerable delay time. Their mechanical design requires a more significant number of components, which will increase the complexity and weight of the MUAV. The induced mechanical constraints and friction leads to a reduction in folding ratio and actuation capability. The associated mechanical complexity in 3D morphing frames could lead to clunky and bulky MUAV, limiting the flight time and loadbearing capacity. It also requires ample space before and after the gap.

Hence, this research can address the above challenges by designing a straightforward morphing approach that allows our MUAV to perpetuate structural simplicity without requiring sophisticated folding mechanisms. Table 1 shows the advantages of the developed morphed MUAV over the conventional MUAVs. The morphing strategy comprehends two elements operating in collaboration: a central frame with four arms that fold around the mainframe and an Arduino microcontroller-based control scheme with a proximity sensor that can sense the presence of obstacles. Our morphing strategy in the framework of rotating links incorporated in the mechanism to be a reasonably good choice in terms of versatility and real-time implementation [8,18]. A rise of singular configuration during morphing is another crucial aspect of selecting the morphing scheme to attain stable flight without control loss. The implemented morphing mechanism should give the flying MUAV dexterity by instantly diminishing its wingspread whilst preserving its high payload carrying capacity. The system does not require any aggressive manoeuvres, but it must be a fast embedded mechanism for folding and deploying MUAVs structure. Furthermore, it should support the implemented autonomous narrow gap crossing strategy based on onboard sensing and computing process. Thus, we adopt a simple, resilient and manipulatable planar folding approach comprising four folding arms to handle this issue. The remaining part of this article is structured as follows; Section 2 and Section 3 describe the mechanical design and kinematics of the morphed configuration of MUAV, respectively. Section 4 provides results of kinematic simulation using Solid works software. Section 5 discusses hardware specification and feature of morphed MUAV, and Section 6 presents experimental validations and discussions to achieve “H-Morphology” with the synchronous motion of the four arms. Finally, Section 7 purveys concluding remarks and future works.

## 2. Mechanical Design 

The foldable MUAV design incorporates a couple of paramount modules: (i) the stiff intermediate frame accommodating the battery, controller and other electronic modules and (ii) quadruple foldable arms with rotors. Each arm has a mutable angle $\beta_{i}\left(i=1,\ldots ,4\right)$ about the body axis regulated by a servo motor, as shown in Figure 1a. The MUAV can change its configuration in flight from the ordinary X-configuration Figure 1a,$\;\beta_{i}=\frac{\pi }{4},\;\left(i=1,\ldots ,4\right)$ to the mission-inherent morphologies while arbitrating the flight time and manoeuvrability. Once the task is completed, the MUAV regains the X-configuration to retrieve its flight performance and manoeuvrability. For instance, in our case, the front and rear arms were folded to traverse through narrow gaps. Accordingly, the MUAV adopts a compact H-configuration as shown in Figure 1b. However, the H-configuration has less manoeuvrability during the roll motion than the standard X morphology; thus, it’s required to revert back into X-configuration after completing the designated task. Each rotating arm is treated as a single degree of freedom (DOF) robotic arm with one revolute joint coupled to the servomotor through a direct drive. It reduces the mechanical complexity of the mechanism while improving its morphing ability to assume various configurations depending on the available gap without needing any additional component.

### 2.1. Kinematic Modelling

#### 2.1.1. Forward Kinematics of the Folding Arm

In order to determine the positional information of the tip of each arm, various Co-ordinate frames are assigned. As it is depicted in Figure 2, a co-ordinate $b_{0}\left[x_{0},y_{0}\right]$ is the fixed frame attached at the origin of the body, $b{\;}_{1}\left[x_{1},\;y_{1}\right]$ represents the frame attached to the standard X configuration of the arm, whereas $b_{1}^{\prime }\;\left[x_{1}^{\prime },\;y_{1}^{\prime }\right]$ represents the rotating frame of $b{\;}_{1}\left[x_{1},\;y_{1}\right]$ by an angle β (i.e., $\beta =0,\ldots \ldots \ldots \ldots ,\frac{\pi }{4}$). The vector $p{\left[x_{1},y_{1}\right]}^{T}$ represents the position of arm with respect to frame $b{\;}_{1}\;$and $p^{\prime }{\left[x_{0},y_{0}\right]}^{T}$ represents the arm’s translated position concerning the body frame $b_{0}$. The arm is rotated with respect to “$Z_{1}\;\mathrm{and}\;Z_{1}^{\prime }$” axis of frame$\;b_{1}\;\mathrm{and}\;b_{1}^{\prime }$ and hence, the rigid point “*P*” attached to the arm translates into a new position “$\;P^{\prime }\;$“ based on the input angle of rotation “*β*” provided by the servo motor as shown in Figure 2. A rotation transformation matrix representing the new position of the rigid point “*P*” concerning the fixed body reference frame $b_{0}$ is formulated. 

Where $L_{arm}$ is the total length of the arm, $W_{b}$ is width of the base frame, *P* is the fixed point attached to the arm and, $P^{\prime }$ is the new translated point after being rotated by an angle β.

The position of a fixed point “*P*” with reference to $“b{\;}_{1}”\;$frame is given by,
(1)$$P\;{\left[x_{1,}^{\prime }y_{1}^{\prime }\right]}^{T}={\left[\;L_{arm}\sin\left(\frac{\pi }{4}\right)-(L_{arm}\cos\left(\frac{\pi }{4}\right))\right]}^{T}$$
where the arm rotates from *P* to $P^{\prime }$ with an angle β is calculated using the following relation:(2)$$P^{\prime }\;{\left[x_{1,\;}^{\prime }y_{1}^{\prime }\;\right]}^{T}=R_{z}\;*\;P\;{\left[x_{1,\;}^{\prime }y_{1}^{\prime }\;\right]}^{T}$$

The position vector after performing the rotation of the arm is given by,
(3)$$P^{\prime }=\left[\begin{array}{c} L_{arm}\sin\left(\frac{\pi }{4}+\beta \right)\\ L_{arm}\cos\left(\frac{\pi }{4}+\beta \right)\\ 0 \end{array}\right]$$

It is evident from Figure 2 that, the two co-ordinate frames $\;b_{0},\;b_{1}^{\prime }$ are parallel. Thus, the position of $P^{\prime }\;$with regard to body-fixed frame $b_{0}$ is obtained by:(4)$$P^{\prime }{\left[x_{0},y_{0},\;z_{0}\right]}^{T}=P^{\prime }\left[x_{1},y_{1},z_{1}\right]+{\left[\frac{L_{b}}{2},\frac{W_{b}}{2}\right]}^{T}=\left[\begin{array}{c} L_{arm}\sin\left(\frac{\pi }{4}+\beta \right)+\frac{L_{b}}{2}\\ L_{arm}\cos\left(\frac{\pi }{4}+\beta \right)+\frac{W_{b}}{2}\\ 0 \end{array}\right]$$

Equation (4) can be used to determine the new translated position of rigid point attached to a rotating arm with respect to the body-fixed reference frame.

##### Modelling the Folding Mechanism

The mechanism encompasses an identical set of bars secured by revolute joints with one DOF, providing a compact and expanded arrangement. The servomotor regulates the horizontal wingspan ($L_{H}$) and vertical wingspan ($L_{V}$) of the arm by adjusting the folding angle β. The morphed MUAV configuration parameters during folding and deploying the arm are shown in Figure 3.

$L_{H}$ denotes the horizontal distance between the two arms, including the radius of the two propellers “*rp*”. When the arm is driven to realize a compact structure the $L_{H}$ value will vary from $L_{H1}$ to $L_{H2}$ as it was depicted in Figure 3. Where $L_{H1}$ is the maximum unfolded horizontal space between the arms and $L_{H2}$ is the minimum folded horizontal space between the arms. $L_{V}$ indicates the total vertical height of the morphological MUAV, including the radius of the two propellers. It also depends on the angular rotation of the arm ranging from $L_{V1}$ (the vertical gap between the foldable arms in the original X configuration) to $L_{V2}$ (the vertical gap between the foldable arms in its confined structure). $W_{b}$ indicates the MUAV body frame’s width. $L_{b}$ denotes the height of the MUAV body frame in which the foldable arms attached. *β* represents the angle of rotation of the arm; it is also known as the folding angle. L is the length of the foldable arm.

The horizontal wingspan of the arm can be calculated using the following relation which is given by:(5)$$L_{H}=2\times (r_{p}+L\;\times \cos(\frac{\pi }{4}+\beta ))+{\;W}_{b}$$

The mechanical performance of the mechanism to traverse through narrow gaps is directly related to a parameter called “folding ratio”, which is determined by the ratio between the enfolded and released horizontal space between the arms. 

Hence, the folding ratio is given by:(6)$$\eta_{H}=\frac{L_{H\;folded}}{L_{H\;unfolded}}=\frac{2\times \left(r_{p}+L\times \cos(\frac{\pi }{4}+\beta )\right)+W_{b}}{2\times \left(r_{p}+L\times \cos(\frac{\pi }{4})\right)+W_{b}}=\frac{2\times \left(r_{p}+L\times \cos(\frac{\pi }{4}+\beta )\right)+W_{b}}{2\times \left(r_{p}+\frac{L\sqrt{2}}{2}+\frac{W_{b}}{2}\right)}$$

From Equation (6), the efficacy vis-à-vis the folding ratio corresponds to geometric constraints, namely the propeller radius, the length of the arms and the base frame’s width. When the MUAV is folded, the horizontal gap is diminished, whereas the vertical length of the MUAV is increased. The vertical wingspan of the arm is given by:(7)$$L_{V}=2\times (r_{p}+L\times \sin(\frac{\pi }{4}+\beta ))+L_{b}$$

The increase in the length can be expressed using the folding ratio for the vertical length, which is given by:(8)$$\eta_{v}=\frac{L_{V\;folded}}{L_{V\;unfolded}}=\frac{2\times \left(r_{p}+L\times \sin(\frac{\pi }{4}+\beta )\right)+L_{b}}{2\times \left(r_{p}+L\times \sin(\frac{\pi }{4})\right)+L_{b}}=\frac{2\times \left(r_{p}+L\times \sin(\frac{\pi }{4}+\beta )\right)+L_{b}}{2\times \left(r_{p}+\frac{L\sqrt{2}}{2}+\frac{L_{b}}{2}\right)}$$

In our model, the fully folded arrangement (i.e., β = 45°) can have a horizontal wingspan equal to 220 mm, which leads to a folding ratio of η = 40%. However, the wingspan for the spread-out arrangement (i.e., β = $0^{{}^{\circ}}$) is equal to 530 mm. In other words, during the folded condition, the MUAV can navigate through a gap two times lesser than its typical wingspan. The same is being applied to the vertical wingspan of the MUAV, and the folding ratio is calculated using Equation (8). Once the arms have enfolded, 25% of the increment in the body’s length is achieved. Table 2 provides the comparison of folding ratio for folded and unfolded arrangement. 

Figure 4 illustrates the change in folding ratio during the actuation of servo motor at different angles (i.e., $\beta =0,\ldots \ldots \ldots \ldots ,\frac{\pi }{4}$) calculated using Equations (6) and (8). It was observed that, initially, the horizontal folding ratio of “X” Morphology was 100 %. However, if the servo motor was driven to achieve the compact structure, the horizontal folding ratio decreases to 40%, enabling 60% reduction in horizontal space. Whereas the vertical folding ratio will increase by 25% when the arm was folded, but after performing the folding task, it is necessary to retrieve the original configuration at this instant, the vertical folding ratio will decrease as depicted in Figure 4b.

## 3. Modeling of Quadrotor UAV

The Quadrotor UAV system has four rotors. The fixed frames of body ($O_{b},X_{b},Y_{b},$ $Z_{b}$) and earth ($O_{e},X_{e},Y_{e}$, $Z_{e}$) are defined. The absolute position of the system is determined by the vector $\xi ={\left[x,y,z\right]}^{T}$. $\eta ={\left[\varphi ,\theta ,\psi \right]}^{T}$signifies Euler angles. $\Omega ={\left[p,q,r\right]}^{T}$and $V={\left[u,v,w\right]}^{T}$ denote the angular and linear velocities, correspondingly. The Newton-Euler equations of Quadrotor model are presented by [34,35].
(9)$$\begin{array}{c} \dot{\xi }=V\\ m\dot{V}=RFe_{3}-mge_{3}-k_{i}\\ \dot{R}=RS\left(\Omega \right),\\ J\dot{\Omega }=-\Omega \times J\Omega +M_{c}-M_{a}+M_{b} \end{array}$$
where *R* is rotation matrix as:(10)$$R=\left[\begin{array}{ccc} C_{\theta }C_{\psi } & S_{\varphi }S_{\theta }C_{\psi }-C_{\theta }S_{\psi } & C_{\varphi }S_{\theta }C_{\psi }+S_{\varphi }S_{\psi }\\ C_{\theta }S_{\psi } & S_{\varphi }C_{\theta }S_{\psi }+C_{\varphi }S\psi & C_{\varphi }S_{\theta }S_{\psi }-S_{\varphi }C\psi \\ -S_{\theta } & C_{\varphi }C_{\theta } & C_{\varphi }C_{\theta } \end{array}\right]$$
and the coordinate transformation is given as:(11)$$T=\left[\begin{array}{ccc} 1 & 0 & -S_{\theta }\\ 0 & C_{\varphi } & S_{\varphi }C_{\theta }\\ 0 & -S_{\varphi } & C_{\varphi }C_{\theta } \end{array}\right]$$
where $C\left(.\right)$ and $S\left(.\right)\;$are cos(.) and sin(.), respectively. m shows the total mass of quadrotor, $I_{3\times 3}$ represents the density matrix, $J=diag\left(I_{x},I_{y},I_{z}\right)$ signifies moment of the inertia. F indicates the total thrust with $F=F_{1}+F_{2}+F_{3}+F_{4}$ where $F_{i}=b\Omega_{i}{}^{2}$, $\Omega_{i}$ represents the angular speed and b denotes the air density and blade geometry parameter. $S={\left[p,q,r\right]}^{T\;}$is a skew-symmetric matrix. $M_{a},M_{c}$ and $M_{a}$ indicate the aerodynamic friction torque, gyroscopic effect torque and four rotors torque, correspondingly, with
(12)$$M_{a}=diad(k_{4}{\dot{\varphi }}^{2},k_{5}{\dot{\theta }}^{2},k_{6}{\dot{\psi }}^{2}),$$
(13)$$M_{c}=\overset{4}{\underset{i=0}{\sum }}\Omega \times J_{r}{\left[0,0,{\left(-1\right)}^{i+1}\Omega_{i}\right]}^{T}$$
(14)$$M_{b}=\left[\begin{array}{c} l\left(F_{3}-F_{1}\right)\\ l\left(F_{4}-F_{2}\right)\\ d\left(-\Omega_{1}+\Omega_{2}-\Omega_{3}+\Omega_{4}\right) \end{array}\right]$$
where $k_{4},k_{5}$ and $k_{6}$ are friction aerodynamics parameters, $J_{r}$ is the rotor inertia, d denote the drag coefficient, l is the distance between a propeller and Quadrotor’s center. The total Quadrotor dynamics with external disturbances is presented by [36,37].
(15)$$\begin{array}{c} \ddot{\varphi }=\dot{\theta }\dot{\psi }\frac{I_{y}-I_{z}}{I_{x}}-\dot{\theta }\frac{J_{r}}{I_{x}}\Omega_{r}-\frac{k_{4}}{I_{x}}{\dot{\varphi }}^{2}+\frac{1}{I_{x}}U_{2}+d_{\varphi },\\ \ddot{\theta }=\dot{\varphi }\dot{\psi }\frac{I_{z}-I_{x}}{I_{y}}+\dot{\varphi }\frac{J_{r}}{I_{y}}\Omega_{r}-\frac{k_{5}}{I_{y}}{\dot{\theta }}^{2}+\frac{1}{I_{y}}U_{3}+d_{\theta },\\ \ddot{\psi }=\dot{\varphi }\dot{\theta }\frac{I_{x}-I_{y}}{I_{z}}-\frac{k_{6}}{I_{z}}{\dot{\psi }}^{2}+\frac{1}{I_{z}}U_{4}+d_{\psi },\\ \ddot{x}=\frac{1}{m}\left(\left(C_{\varphi }S_{\theta }C_{\psi }+S_{\varphi }S_{\psi }\right)U_{1}+d_{x}\right)-\frac{k_{1}}{m}\dot{x},\\ \ddot{y}=\frac{1}{m}\left(\left(C_{\varphi }S_{\theta }S_{\psi }-S_{\varphi }C_{\psi }\right)U_{1}+d_{y}\right)-\frac{k_{2}}{m}\dot{y},\\ \ddot{z}=\frac{1}{m}\left(C_{\varphi }C_{\theta }U_{1}+d_{z}\right)-g-\frac{k_{3}}{m}\dot{z} \end{array}$$

The Quadrotor inputs and propeller speeds are related to each other in the following form:(16)$$\left[\begin{array}{c} U_{1}\\ U_{2}\\ U_{3}\\ U_{4} \end{array}\right]=\left[\begin{array}{cccc} b & b & b & b\\ -lb & 0 & lb & 0\\ 0 & -lb & 0 & lb\\ -d & +d & -d & d \end{array}\right]\left[\begin{array}{c} \Omega^{2}{}_{1}\\ \Omega^{2}{}_{2}\\ \Omega^{2}{}_{3}\\ \Omega^{2}{}_{4} \end{array}\right]$$

The dynamics of position subsystem of Quadrotor has three outputs, i.e., z,x, y and one controller $U_{1}$. The virtual controllers are provided as.
(17)$$\left[\begin{array}{c} v_{1}\\ v_{2}\\ v_{3} \end{array}\right]=\left[\begin{array}{c} \frac{U_{1}}{m}\left(C_{\varphi }S_{\theta }C_{\psi }+S_{\varphi }S_{\psi }\right)\\ \frac{U_{1}}{m}\left(C_{\varphi }S_{\theta }S_{\psi }-S_{\varphi }C_{\psi }\right)\\ \frac{U_{1}}{m}\left(C_{\varphi }C_{\theta }\right)-g \end{array}\right]$$

Hence, the desired roll, pitch and yaw angles ($\varphi_{d},\theta_{d}$) and the thrust controller $U_{1}$ are: (18)$$\begin{array}{c} U_{1}=m\sqrt{v_{1}{}^{2}+v_{2}{}^{2}+{\left(v_{3}+g\right)}^{2}},\\ \varphi_{d}=\arctan\left[S_{\theta d}\left(\frac{v_{1}C_{\psi d}-v_{2}S_{\psi d}}{v_{3}+g}\right)\right],\;\\ \theta_{d}=\arctan\left(\frac{v_{1}C_{\psi d}+v_{2}S_{\psi d}}{v_{3}+g}\right). \end{array}$$

## 4. Simulation Analysis of Morphed Configuration

The simulation was performed using SOLIDWORKS (2018) software platform to analyze and validate the derived kinematics relationships. In order to provide a valid input for simulation, a “TowerPro MG996R (Bombay Electronics, Mumbai, India)” digital high torque metal gear servo motor data was utilized, and its specification is given in Table 3. It can be seen that under 6V supply, the TowerPro MG996R servo provides higher torque and angular velocity features in comparison to the value under 4.8V. 

A set of simulation frames in (Figure 5) shows the folding and unfolding sequences of MUAV. The results of simulation and analysis during these configurations are shown in Figure 6. The simulation events A, B, C and D indicates the accompanying formations when the Quad structure was switched. The moment A = 0 s specifies the static unfolded formation, whereas B = 0.5 s designates the change of the formation from unfolded to folded. The moment C = 1 s and D = 1.5 s specify the static folded formation and change of the formation from folded to unfolded. 

The folding succession was achieved from A to C, and the unfolding succession was attained from C to A. Point B is the instant when the arm is folding, and D is the instant when the arm is unfolding. It can be seen from Figure 6a that the motor can rotate the arm $45^{{}^{\circ}}$ from its initial position, which is the desirable pose transformation of the motor arms to prevent collision of propellers with each other, ensuring a 60% reduction in volume.

The angular velocity of the MUAV arm during folding and unfolding is shown in Figure 6b. It can be observed that $\dot{\beta }$ manifests mainly linear character signifying a uniform acceleration motion. It was evident from the simulation analysis that the required angular velocity to drive the arm was found to be 100 deg/s. However, for the given input of 4.8 V, the servo motor can rotate with an angular velocity of 315.8 deg/s which means, the selected servo motor can perform rotation efficiently.

The torque requirement to actuate the metal gear servo was also examined. It was evident from Figure 6c that, the maximum torque required was estimated as 0.2 Nm which is less than the stall torque of the considered MG996R metal gear servo (Bombay Electronics, Mumbai, India) under 4.8 V.

## 5. Hardware Specification and Construction of Morphed MUAV

The reduction of workspace volume of MUAV was considered to be the primary objective for accomplishing the morphed configuration to navigate into the obstacle prone regions. The morphed MUAV has to be designed for its affordability, easily reconfigurable, robust and high mechanical performance. The hardware components such as motors, propellers, electronic speed controllers and frames were selected to provide such characteristics. The specifications and hardware components of the morphed MUAV are listed in Table 4. 

The custom-built morphed MUAV is shown in Figure 7 wherein all the electronic modulus listed in Table 4 were embedded at the central frame of MUAV. The MUAV structure weighs 1.5 kg, and it can take up a payload of 0.5–1 kg by compromising the flight time.

## 6. Experimental Analysis

Various experiments were performed to validate the conceptual design of the synchronous folding and unfolding of the MUAV arm. The first experiment was conducted to measure the actual elapsed time while simultaneously folding and unfolding each of the four arms. It was conducted to assess the possibility of achieving the necessary volume reduction within a defined period following intermediate configuration modifications in real-time, and the experimental results were compared to the simulation results. In the second experiment, the battery discharge rate was measured in hovering flight mode to verify the effectiveness of the designed mechanism. Based on the data logs obtained from the onboard autopilot system, the battery performance was measured and transmitted to the mission planning software through the (Micro Air Vehicle Link) MAVLink telemetry. Finally, it was essential to test its operation under different actuation speeds by incorporating an obstacle avoidance algorithm to avoid obstacles in real-time. For this experiment, the proposed obstacle avoidance algorithm was tested using the Arduino Nano platform. It supports the serial monitor display, which shows the angle travelled by the arm at each instant of time based on the measured distance from the ultrasonic sensor. A WiFi module was utilised to connect the Arduino to the serial monitor. In addition, the radial vibration-induced during the flight was also obtained from the flight log data on the mission planner software. The results revealed that the MUAV had considerable vibration; therefore, it was damped with a damper and kept within a safe limit.

### 6.1. Synchronous Folding and Un-Folding 

During this experiment, the actual time required for the MUAV to accomplish a fully folded and unfolded configuration was measured, and the comparison was made with the simulation results. 

Hardware testing of the developed morphed UAV configuration through actuating the servomechanism to achieve the folding of UAV structure for reducing the work volume was performed. From Figure 8 it was observed that the MUAV structure switches from un-folding to folding (A–C) and folding to un-folding (C–A). The trajectory during the actuation event was almost similar to the results obtained through simulation and analysis except for few positional errors due to certain mechanical constraints. In addition, it can be inferred that when the arm attains a compact configuration, the horizontal folding ratio was reduced to 40%, enabling a 60% reduction in horizontal space, whereas the vertical folding ratio was increased by 25%. However, after performing the folding task, it necessitates retrieving back to the original configuration so that the Morphed MUAV regains its nominal flight efficiency and performance.

The experimental results have clearly shown that the added folding mechanism opposes no impediment to the operation of morphed MUAVs in real-time and advocates a significant decrement in structural volume. The time elapsed when the arms were folding and unfolding was demonstrated in Figure 9. It can be observed that the MUAV can change from the unfolded to folded state (and vice-versa) in approximately one second which was observed in the simulation results as well. In addition, due to the rigidity of the motor arm, the durability of the folding and unfolding mechanism was enhanced, which improves the efficiency of operation. 

### 6.2. Power Consumption and Flight Endurance

The flight time (endurance) is an essential characteristic of any MUAV configuration, and it indicates that the design of MUAV and also the selection of hardware components is appropriate without significant loss in power. The endurance of MUAV depends on the power consumption of various electronic modules and Brushless Direct Current (BLDC) motors. The flight time was calculated by using the morphed UAV parameters given in Table 4. It was observed that the total available energy was 76.96 Wh for the 5700 mAh battery capacity. Basic experiments were conducted in indoor conditions to measure the power consumption of all electronic elements. The BLDC motors were drawn approximately 160 W and other electronic modules except the component’s used to trigger the arm consumed 14.5 W power during hovering flight. Thus, the average power consumption of the whole system of the conventional MUAV is 174.5 W. Hence, the flight duration for hovering mode was calculated as;
$$Flight\;Time=\frac{76.96\;\mathrm{Wh}}{174.5\;W}=0.44\;H=27\;min$$

Considering the morphed MUAV based on crank-rocker mechanism as it was presented in [13], the additional sensing and actuation modules for the folding mechanism consume approximately 11.1 W power. However, the custom developed morphed MUAV based on rotating links consumed only 6 W to actuate each of the four arm synchronously. The flight endurance was determined for both cases considering the power consumed to actuate each of the arms and the results obtained were shown in Table 5. 

Compared with the basic MUAV configuration, developed planar folding strategy based on rotating links achieved 1 min decrement in the flight endurance of MUAV which was very minimal for deploying the morphed configuration in real time without compromising the endurance. The battery discharge rate in hovering flight mode for the whole system of the developed morphed MUAV is shown in Figure 10. Moreover, it can be inferred that there was no drastic change in battery consumption, so the morphed MUAV was able to achieve steady flight for almost 26 min as it was calculated. The discrepancy between the calculated flight time and actual flight time can be elucidated by factors such as environmental conditions, altitude density, drag, etc. 

Figure 10 shows the battery performance with respect to the maximum available voltage of cells. It was evident that the morphed MUAV can achieve the flight endurance of 24 min which is equal to the calculated endurance. 

### 6.3. Experimental Testing of the Morphing Phenomenon Using Obstacle Avoidance Algorithm

Real-time experiments were conducted using the developed obstacle avoidance algorithm for testing the performance of MUAV. The servo motors were triggered synchronously by using an ultrasonic sensor to detect the presence of obstacles. The sensor sends a signal to the Arduino microcontroller to command the servomotors to change the Quad structure’s morphology. For validating the compatibility of the mechanism, two experiments were conducted. In the first experiment, the obstacle was kept at a distance of 40 mm, and the servo motors were commanded to move 90°/s. The Figure 11a shows the system’s response wherein the input signal was given as the distance from the obstacle and output was the servo rotation angle. In addition, the distance measured by the ultrasonic sensor and the actual distance from the obstacle was compared, and it has achieved 99% of accuracy. It was evident that the servomotors can drive the arms to the destination within a second as expected, even if there were inconsistencies in the ultrasonic sensor due to environmental disturbances and other uncontrolled factors. In the second experiment, the obstacle was placed at 120 mm as shown in Figure 11b. In this experiment, the servos were commanded to move 45° in two seconds. It was evident from the Figure 11 that, the 2Dmorphing strategies based on the developed mechanism assembly attained faster actuation of the arm when the obstacle was closer to the system. On contrary, it was slower and smoother actuation when the obstacle was far from the sensor. These higher range in speed operations were not observed in the earlier works because their mechanical design requires many components, increasing the MUAV’s complexity and weight. The induced mechanical constraint and friction results in a more extended time for altering their configuration. It leads to a considerable delay in execution time and thus it may not perform swift action to navigate into the narrow environment. However, the developed morphed MUAV configuration can navigate into the confined regions swiftly and reconfigure it with appropriately.

### 6.4. Vibration Measurement

In the event of real-time flight conditions of MUAV, the structure’s vibration significantly affects the stability. In order to measure the vibration of the developed MUAV structure, experiments were conducted inside a cage to eliminate the interference of the signal. Initially, without a damper the MUAV experiences high vibration and by inserting a rubber damper at the bottom of BLDC motors the amplitude of vibration was reduced. The experimental results depicted in Figure 12 show that the radial vibrations acting on the cartesian body frame of MUAV was modest. It was also observed that the oscillation was between ±0.05 g, which was lower than the expected interference that may cause a threat to the stability of MUAV while adapting different morphologies in flight.

## 7. Conclusions and Future Work 

The present work focused on the development of morphed MUAV for achieving effective navigation in obstacle prone environments. The mechanical design and corresponding kinematic analysis results suggested that the 45° rotation of MUAV arms resulted in a 60% reduction in the horizontal workspace and a 25% increase in vertical workspace volume, which was very much needed to fly the MUAV in cluttered regions. The simulation results also suggested that 0.2 Nm torque was required to actuate the servo motor to overcome the self-weight and inertia of the MUAV arms, and a corresponding digital high torque metal gear servo motor was selected. Experimental results confirm that the developed morphed MUAV at hovering flight achieved 26 min flight endurance. The flight trials revealed that the morphed MUAV could efficiently alter its configuration from “X” to “H” shape in real-time during the presence of obstacles and reassume its original configuration. The synchronous motion of the four arms was achieved without time delay, and the vibration of the entire structure was minimized by employing foam tape beneath the autopilot system. The developed morphed MUAV can be well suited for performing proficient navigation in the clustered environment to complete the designated tasks effectively through changing its configuration, which the conventional MUAVs may not accomplish.

Finally, the foremost contribution of our paper is an entirely autonomous and mechanically compact work-volume reduction strategy for MUAVs, which alleviates the challenge of space limitation while it operates in cluttered environments. In the future, integration of the developed morphing approach with advanced control algorithms to improve the MUAVs resilience while assuming different morphologies during flight will be performed. Furthermore, an artificial intelligence-based system will be developed to achieve multiple morphological configurations of MUAVs to traverse through confined spaces in dense environments, such as in disaster-prone regions.

## Figures and Tables

**Figure 1 sensors-21-06192-f001:**
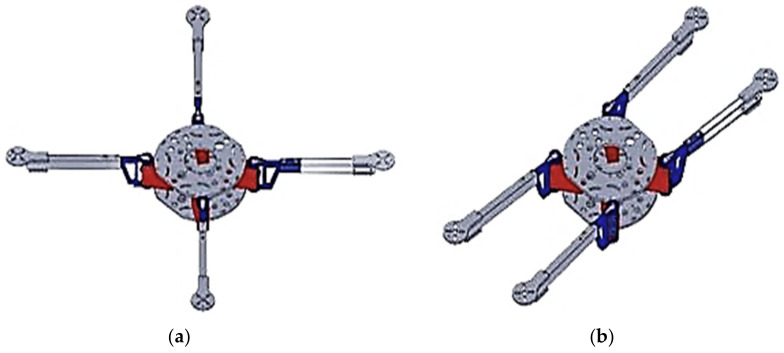
Configurations of MUAV (**a**) The unfolded “X” configuration of MUAV. (**b**) The folded “H” configuration of MUAV.

**Figure 2 sensors-21-06192-f002:**
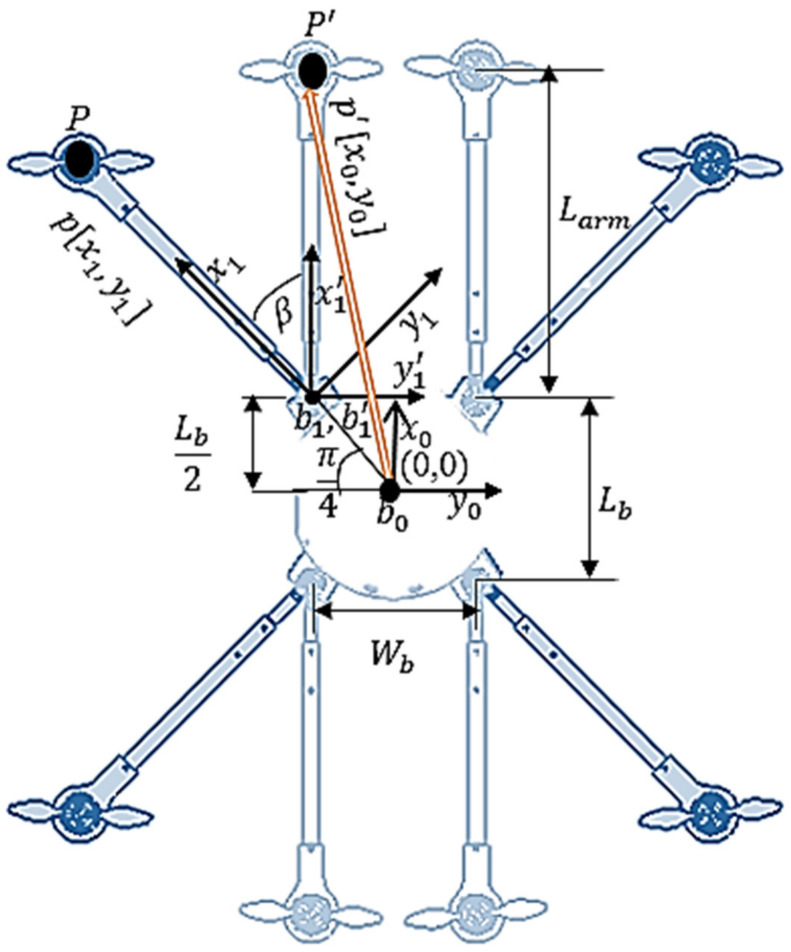
Morphed MUAV with Co-ordinate Frames.

**Figure 3 sensors-21-06192-f003:**
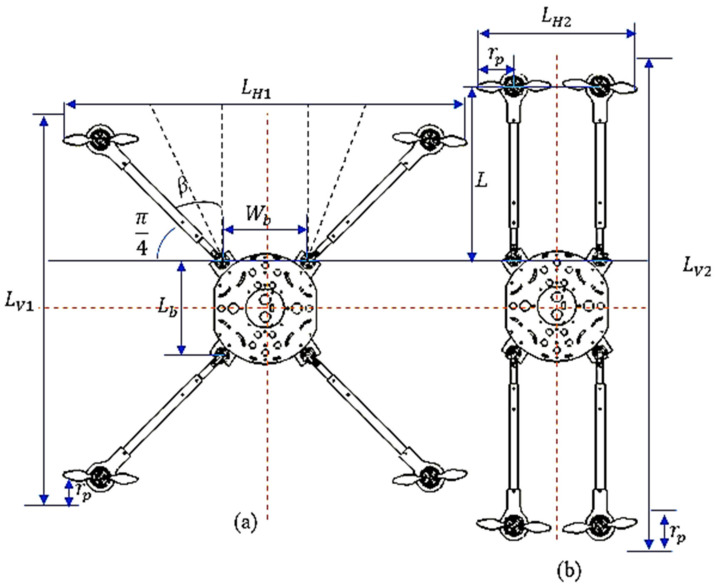
Geometry of the morphed MUAV: (**a**) unfolded MUAV in “X” Configuration (**b**) folded MUAV in “H” Configuration.

**Figure 4 sensors-21-06192-f004:**
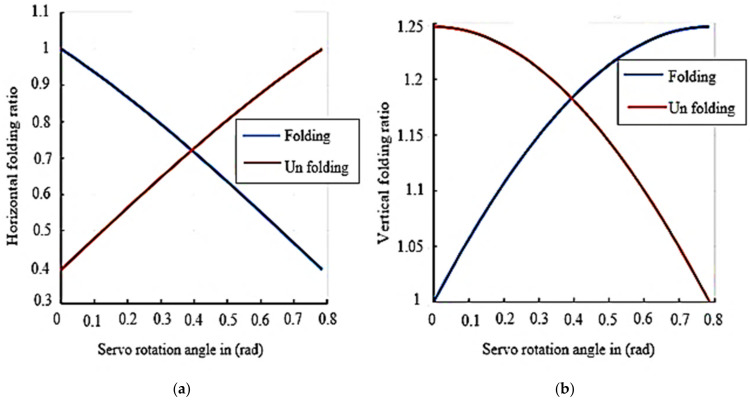
Folding ratio (**a**) Horizontal folding. (**b**) Vertical folding.

**Figure 5 sensors-21-06192-f005:**
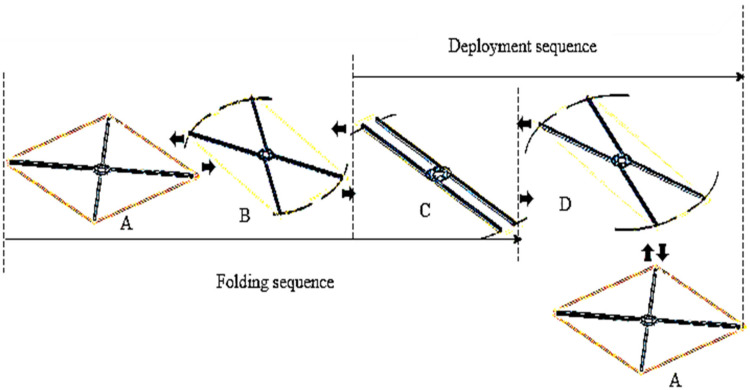
Folding and unfolding sequence of the MUAV arms.

**Figure 6 sensors-21-06192-f006:**
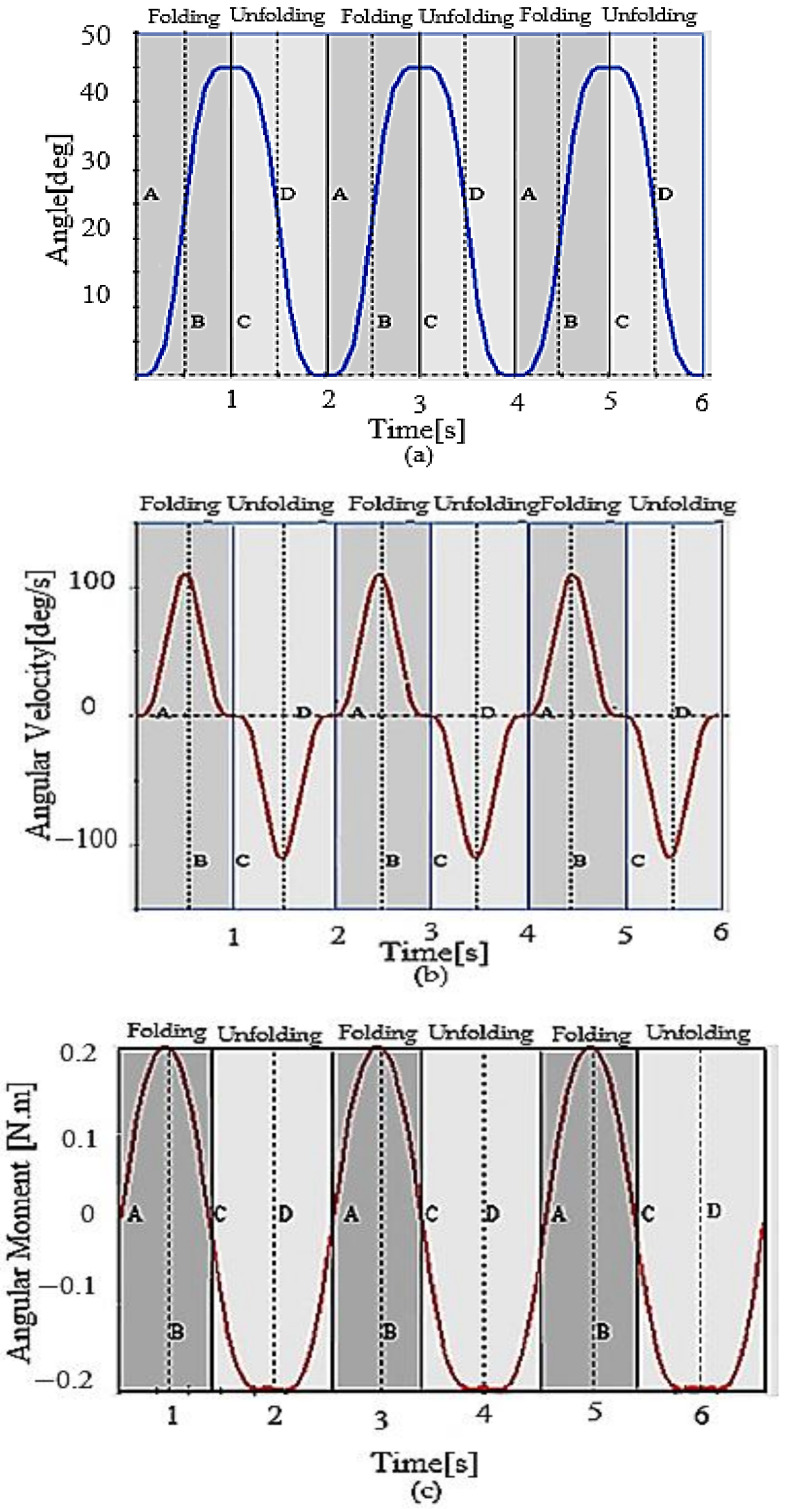
Analysis of the folding mechanism; (**a**) Angular position of the motor arm (**b**) The required angular velocity for servo motor actuation (**c**) The required torque for servo motor actuation.

**Figure 7 sensors-21-06192-f007:**
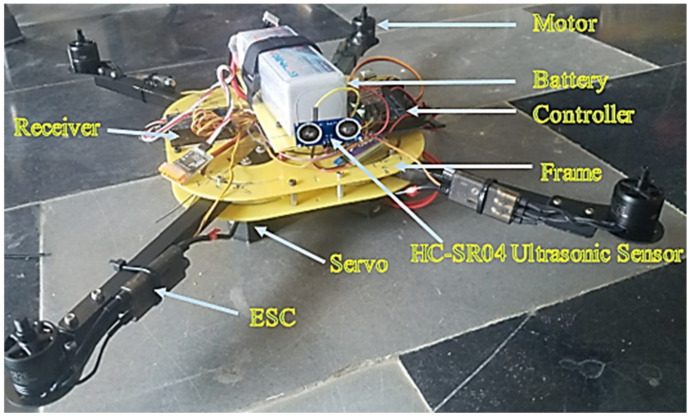
Developed MUAV.

**Figure 8 sensors-21-06192-f008:**
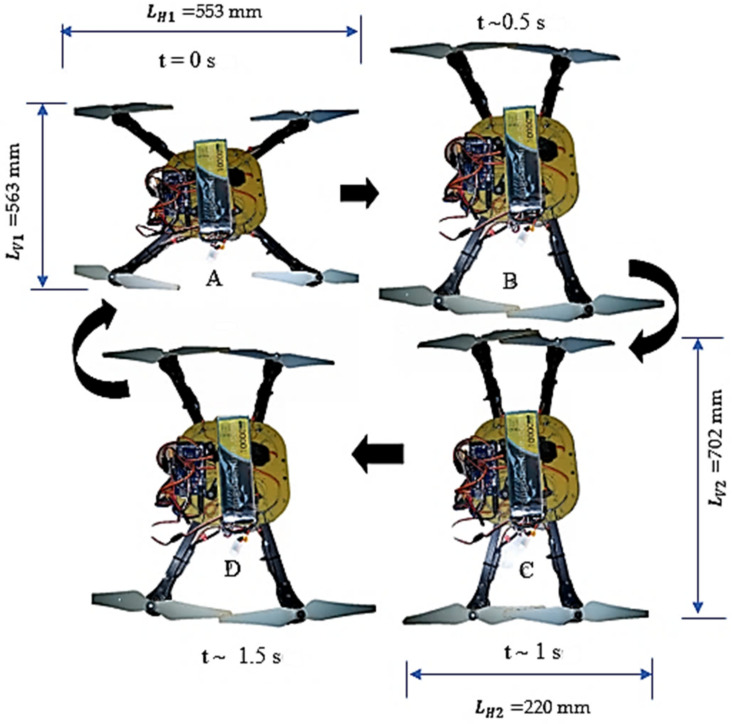
Unfolded and folded configuration of MUAV.

**Figure 9 sensors-21-06192-f009:**
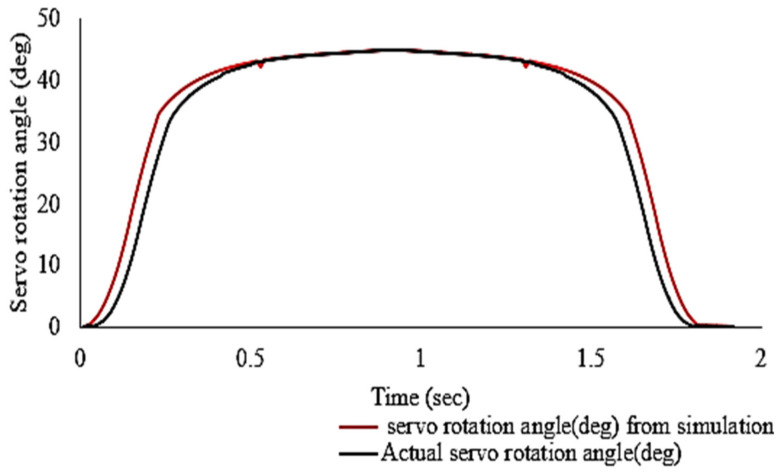
Comparison between experimental and simulation results.

**Figure 10 sensors-21-06192-f010:**
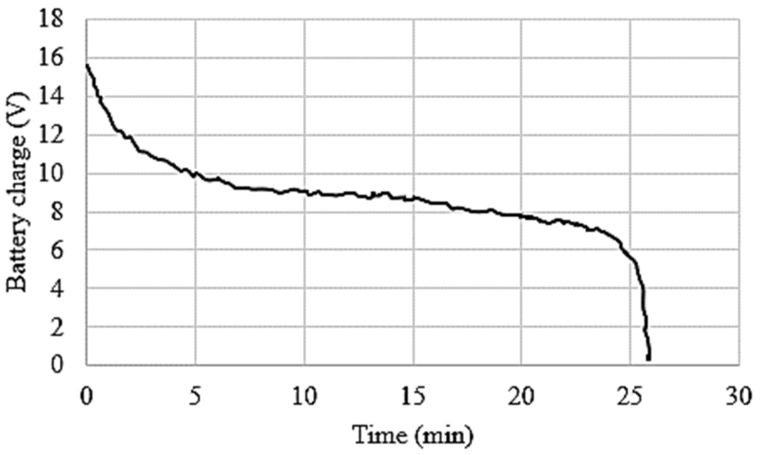
Endurance of MUAV.

**Figure 11 sensors-21-06192-f011:**
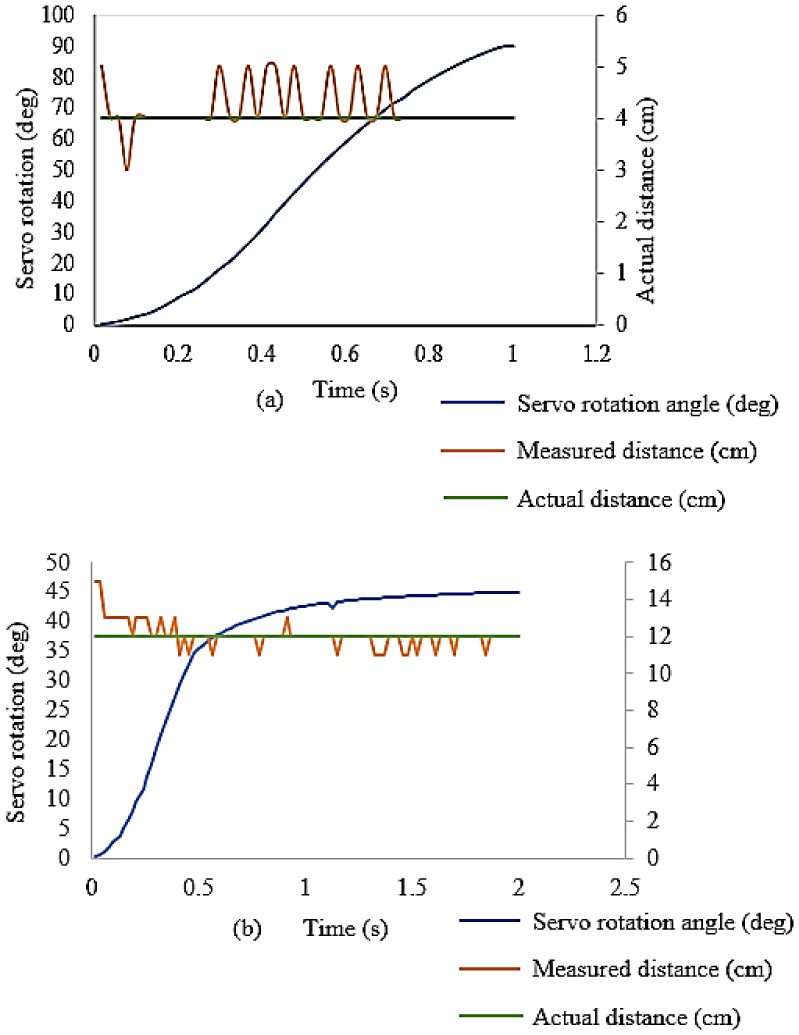
Ultrasonic sensors response and servo motor actuation (**a**) Sensor kept at 40 mm (**b**) Sensor kept at 120 mm.

**Figure 12 sensors-21-06192-f012:**
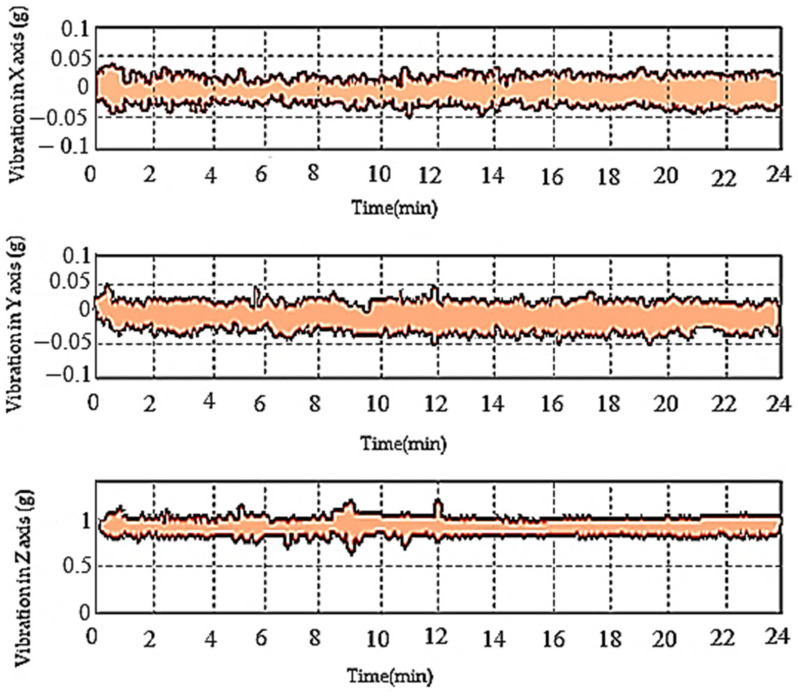
The vibration of MUAV during hover flight.

**Table 1 sensors-21-06192-t001:** Comparison of the developed morphed MUAV with conventional morphed MUAVs.

Comparison Criteria	Developed Morphed MUAV	Custom MUAV
Percentage Reduction in volume	60% reduction in Horizontal space	No space reduction
Obstacle avoidance	Yes	No
Reconfigurable	Yes	No
Intrusion in narrow environment	Yes	No
Adapt to different tasks or environments by altering their shapes	Yes	No

**Table 2 sensors-21-06192-t002:** The comparison of folding ratio for folded and unfolded configuration.

Configuration	Arm Length		Folding Ratio	
	*L_H_* $\left(Horizontal\;Wing\;Span\right)$	*L_V_* $\left(Vertical\;Wing\;Span\right)$	η_H_ $\left(Horizontal\;Folding\;Ratio\right)$	η_V_ $\left(Vertical\;Folding\;Ratio\right)$
Unfolded (β = 0)	553 mm	563 mm	100%	100%
Folded ($\beta =\frac{\pi }{4}$)	220 mm	702 mm	40%	125%

**Table 3 sensors-21-06192-t003:** Specification of TowerPro MG996R digital high torque metal gear servo motor.

Characteristics	Value	
Stall Torque (Nm)	0.92 (@4.8 V)	1.078 (@6 V)
Maximum Idle Angular Velocity (deg/s)	315.8 (@4.8 V)	400 (@6 V)
Instantaneous Current Drawn (mA)	100	120

**Table 4 sensors-21-06192-t004:** Specification of Morphed MUAV.

Weight	1.5 kg
Arm length	220 mm
Propellor diameter	101.6 mm
Object detection	Front
Object detection range	20 mm–4000 mm
Flight time	26 min
Actuators	2300 kv Brushless DC Motors ESC 30 A TowerPro MG996R gear servo motor
Sensors	HC-SR04 Ultrasonic Sensor 20 mm–4000 mm QMC5883L Magnetometer Ublox NEO-M8 Global Positioning System MPU6000 6 Axis SPI Gyroscope + Accelerometer BMP280 Barometer
Controller	Arduino Nano Atmega358–3.3 V/16 MHz Pixhawk PX4 Flight Controller
Power	5700 mAh 11.1 V Li-PO Battery Energy ~ 76.9 Wh

**Table 5 sensors-21-06192-t005:** Flight endurance comparison between existing morphed MUAV and developed morphed MUAV.

Sl. No	Parameters	Developed Morphed UAV	Existing Morphed UAV
1	Power consumption	180.5 W	185.7 W
2	Endurance (5700 mAh)	26 min	24 min

## Data Availability

The data that support the findings of this study are available within the article.

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
