# Peer review of "Design and Implementation of Morphed Multi-Rotor Vehicles with Real-Time Obstacle Detection and Sensing System"

_sensors, 2021, doi:10.3390/s21186192_

Round 1

Reviewer 1 Report

This research paper presents the development of a morphed multi-rotor unmanned aerial vehicle based on a simple rotary actuation mechanism. The paper also presents the mathematical modeling formulation for the folding mechanism along with the corresponding kinematic analysis to understand the synchronous motion characteristics of the arms during the folding of arms. The content of the manuscript is well-organized and well-articulated. However, I feel that the figures and the tables need to be improved to a greater extent. The figures and tables should have a detailed description of the labels and symbols used as it would be easier for the reader to understand. A fair review of the literature has been carried out by the authors. I would suggest adding more literature to the manuscript. The research topic is of interest to the journal and related readers but the use of the English language is not up to the journal standards. Extensive editing of the English language and style is required. The description of how the experiment was conducted and how the analysis was performed should be in past tense and not present tense. I have highlighted some minor comments related to the language and content of the paper.

Author Response

Dear Reviewer

We have addressed all the comments and attached the response. 

Thank you

Reviewer 2 Report

-- 1. Introduction. The section should be more concise and informative. More background knowledge was given while the novelties of the study was not fully introduced. Some quantitative expressions should be provided to describe the advantages of the developed morphed MUAV than conventional MUAVs.

-- When citing the references, it is not necessary to give the full name of the first author and the family name is just ok. For example, Yang et al. [19] is used instead of Dangli Yang et al. [19].

-- The resolutions of most pictures are low and should be improved.

-- There are some variables in Figure 2 and Figure 3, but the explanations are missing. A legend is required to explain the variables in the figures.

-- Figures 6-8 can be merged to a figure and each of them is marked using (a), (b) and (c).

-- Table 3 and Table 4 can be also merged.

-- More explanations of Figure 10 should be given. What is the primary objective to describe the process (from A to D) for the unfolded and folded configuration of MUAV.

-- Some discussions should be in the Section 6. Experimental Analysis. The performance of the developed MUAV should be compared with the previous studies regarding 6.2 Power consumption and flight endurance and 6.3 Experimental testing of the morphing phenomenon using obstacle avoidance algorithm.

Author Response

(The authors gave the same response as above.)

Round 2

Reviewer 1 Report

The suggested changes were made by the authors. The received feedback from the first round of revisions has been incorporated into the paper. The figures and tables have been described in a better way in the revised version. More literature has been added to the journal article and the language has been improved. The revised version of the manuscript is better organized. The research topic is of interest to the journal and the related users. 

Author Response

Dear Reviewer 

Thank you so much for accepting our manuscript

Reviewer 2 Report

Most of the comments raised in the first round have been addressed. There are still some minor corrections that need to be made.

-- The first letter of “sensing” should be capitalized in the title.

-- The citation format of references [1, 2, 3,4] should be changed to [1-4].

-- 1. Introduction. More references especially in the 3-5 paragraphs were just cited and listed to introduce the development of MUAVs, but the literature review was not enough to logically analyze them.

-- The resolutions of figures are still required to be improved.

-- Some discussions can be added in the Section 6. Experimental Analysis.

-- More descriptions should be added for the future work.

Author Response

Dear Reviewer

Greetings!!!

We have incorporated all the suggestions and comments in the revised manuscript. We have attached a response file for addressing all the comments.

Please let us know if there are any other comments. We will be happy to address it.

Thank you so much for your comments to improve our manuscript.
